# Cardiac Surgery in Advanced Heart Failure

**DOI:** 10.3390/jcm11030773

**Published:** 2022-01-31

**Authors:** Roger Hullin, Philippe Meyer, Patrick Yerly, Matthias Kirsch

**Affiliations:** 1Cardiology, Cardiovascular Department, University Hospital and University of Lausanne, Rue du Bugnon 46, 1011 Lausanne, Switzerland; patrick.yerly@chuv.ch; 2Cardiology, Department of Medical Specialties, Geneva University Hospital, University of Geneva, Rue du Gabrielle Perret-Gentil 4, 1205 Geneva, Switzerland; philippe.meyer@hcuge.ch; 3Cardiac Surgery, Cardiovascular Department, University Hospital and University of Lausanne, Rue du Bugnon 46, 1011 Lausanne, Switzerland; matthias.kirsch@chuv.ch

**Keywords:** end-stage heart failure, cardiac surgery, outcome

## Abstract

Mechanical circulatory support and heart transplantation are established surgical options for treatment of advanced heart failure. Since the prevalence of advanced heart failure is progressively increasing, there is a clear need to treat more patients with mechanical circulatory support and to increase the number of heart transplantations. This narrative review summarizes recent progress in surgical treatment options of advanced heart failure and proposes an algorithm for treatment of the advanced heart failure patient at >65 years of age.

## 1. Introduction

Survival with heart failure (HF) has significantly increased in the last decades with the broad implementation of guidelines-directed medical therapy [1]. Nonetheless, HF almost always progresses towards advanced-stage HF characterized by persistent symptoms despite optimal medical treatment. Community-based studies report that advanced HF affects between 1 to 10% of all HF patients [2], and this prevalence will further increase. The reason for this continued surge is the ageing of the population, which involves an exponential rise in the incidence of HF with increasing age [3]. Furthermore, modern HF treatment attenuates progression of HF disease and prolongs survival with heart failure, and this beneficial effect will add further to the burden of advanced HF in the upcoming years [4]. Together, this concomitance explains why patients with worsening HF but also patients with de novo HF are older [5,6,7], which mandates the development of an algorithm taking care of the old patient with advanced HF.

The prognosis of advanced-stage HF remains poor with a 1-year mortality ranging from 25% to 75% [5,6,7]. Often, short-term therapy with inotropes remains the mainstay to improve the clinical condition and to reverse worsening end-organ function despite the fact that this treatment does not improve cardiovascular outcomes and even may worsen the prognosis [8]. Inotropes have therefore no place for routine treatment of chronic refractory advanced HF symptoms while being necessary in the acute setting and acceptable in a palliative-care setting.

In contrast, temporary MCS treatment provides an option to bridge patients with refractory symptoms to long-term mechanical circulatory support (LT-MCS) or heart transplantation (HTx), which remain the two acknowledged options of surgical treatment of advanced HF. While HTx can still be considered as the gold-standard treatment of advanced HF, organ-donor shortages limit the broad application of this treatment option. Whether application of less-stringent criteria for donor selection can increase and expand the donor pool without repercussion on posttransplant survival remains to be followed. On the other hand, LT-MCS therapy has seen a significant surge of activity in the last years, in particular since smaller blood pumps became available [9]. Since then, not only the number of LT-MCS implantation has largely increased but also the clinical experience with this form of surgical treatment of advanced heart failure has expanded significantly. This resulted in an enlargement of the target population of LT-MCS, which nowadays also includes patients with pediatric or older age, congenital heart defects, and more advanced comorbidity.

This review intends to discuss the current evidence for temporary as well as durable MCS treatment and options to expand the donor heart pool for transplantation, and finally, it proposes an algorithm of how to care for the older-age patient presenting with advanced HF.

## 2. Definition of Advanced Heart Failure

Acknowledging that evidence guiding therapeutic decision making in patients with advanced HF remains scarce [4], the European Heart Failure Association of the European Society of Cardiology more recently published a modern definition. This definition revisits existing criteria characterizing this patient group on the one hand while integrating novel elements into the definition on the other hand. Central criteria of this new definition remain the persistent severe symptoms of HF and a reduced maximal exercise capacity. The new definition now acknowledges in addition HF with mildly reduced LVEF and preserved LVEF as an additional cause of advanced HF besides non-operable severe valvular or congenital abnormality, isolated right-ventricular failure, or severe left-ventricular systolic dysfunction. Furthermore, malignant arrythmia is now considered as a clinical event in addition to intermittent episodes of pulmonary or systemic congestion or episodes of low-output requiring an unplanned visit or hospitalization within the last 12 months [8].

## 3. Temporary Mechanical Circulatory Support

The case against broad application of traditional inotropes is their potential to increase myocardial ischemia and to provoke tachyarrhythmia, while inotropes improve central and peripheral hemodynamics, reduce congestion, and alleviate end-organ dysfunction on the other hand [10,11]. Temporary MCS treatment, however, is shown to successfully bridge advanced HF patients to urgent HTx with compelling posttransplant results [12] and, therefore, represents a pertinent treatment option in situations of severe hemodynamic compromise.

Temporary MCS can be provided as bridge to recovery (BTR) or bridge to the decision (BTD) if it is not clear whether the clinical situation may offer future eligibility for durable treatment such as long-term MCS (LT-MCS) or HTx. If the patient is not eligible for either option because of ongoing severe infection, uncontrolled bleeding, or larger dekubitus ulcer, temporary MCS treatment can be maintained until the patient is recovering from temporary contraindication (bridge to candidacy = BTC). However, if the patient is eligible for HTx, temporary MCS treatment can be maintained as a bridge to urgent HTx (bridge to transplant = BTT) [12,13]. Additionally, if the patient is eligible for long-term mechanical circulatory support (LT-MCS) such as destination therapy (DT), temporary MCS may be maintained until clinical improvement provides acceptable peri-operative risk for LT-MCS implantation [14]. Often, these cases are difficult to manage; therefore, discussion of the patient with an advanced heart failure center is recommended to select the most appropriate short-term management strategy [8].

More recent studies provide helpful evidence for decision-making in advanced HF affording temporary MCS treatment and will be therefore discussed in the following.

### 3.1. New Evidence for the Intra-Aortic Balloon Pump

The intra-aortic balloon pump (IABP) is a catheter-mounted balloon sitting in the descending aorta. It augments the pulsatile blood flow by inflation during diastole, thereby displacing blood volume to the distal aorta. Simultaneously, inflation displaces blood to the proximal aorta, increasing thereby the mean pressure in the ascending aorta and the coronary arteries. On the other hand, deflation during systole reduces the afterload [15]. Intuitively, these characteristics should be beneficial to patients with left-ventricular (LV) ischemia and dysfunction; however, large, randomized trials evaluating IABP treatment failed to show improvement in cardiovascular-outcome parameters whether in the setting of acute myocardial infarction with or without cardiogenic shock or in the context of elective high-risk percutaneous coronary intervention [16,17,18,19]. Nonetheless, IABP is still broadly applied in this clinical setting [20].

However, IABP treatment has been applied in recent times more frequently for treatment of advanced HF, especially among patients awaiting HTx. Initially, this application was likely a response to the 2018 United Network for Organ Sharing (UNOS) policy change, which prioritizes patients to status 2 when bridged to heart transplantation with IABP [21]. Surprisingly, waitlist outcomes of these HTx candidates were improved as shown in a recent analysis, and, furthermore, posttransplant survival was not worse in patients with IABP treatment pretransplant [21]. However, prolonged IABP therapy debilitates the physical condition when applied using a transfemoral approach. Therefore, a more recent study tested the feasibility of percutaneous IABP placement via the axillary artery in order to promote ambulation and, thus, mitigate physical debilitation [22]. In this study, 68% of all study participants cases (n = 133) underwent either successful cardiac replacement (n = 115) or left-ventricular assist-device implantation (n = 18) after a median time on IABP support of 19 days. With the longest duration of support being 169 days, this report demonstrates that IABP can be used for an extended duration in patients awaiting HTx. However, IABP needed frequent repositioning, and 37% of study participants required replacement of the IABP due to malfunctioning. In addition, IABP support may not be helpful in advanced HF with important right-heart dysfunction as suggested by increased mortality when the ratio of atrial/pulmonary wedge pressure is high [22].

### 3.2. IMPELLA

The IMPELLA device received in 2016 Food and Drug Administration approval for use in acute myocardial infarction complicated by cardiogenic shock, but its effect on clinical outcomes has remained controversial [23]. However, IMPELLA treatment was associated with a lower mortality when compared to ECMO support in a recent retrospective propensity-matched cohort study of 6290 patients undergoing percutaneous coronary intervention for acute myocardial infarction complicated by cardiogenic shock [24]. This observation may relate to the fact that the IMPELLA increases myocardial perfusion and unloads the left ventricle, whereas the VA-ECMO increases the left ventricular afterload and, thus, may increase the infarct size and worsen clinical outcomes [25,26,27].

For the latter reason, the IMPELLA has been increasingly used as a temporary bridge among decompensated advanced HF patients awaiting heart transplantation [28]. However, it remains unknown whether IMPELLA treatment favorably affects outcomes in this setting.

### 3.3. Venoarterial Extracorporeal Membrane Oxygenation (VA-ECMO)

VA-ECMO remains the most utilized option for temporary MCS in patients with cardiogenic shock. It is a low-cost percutaneous device that allows for rapid implantation and provides full biventricular and pulmonary support. VA-ECMO increases the LV afterload, results in LV dilatation, and augments the incidence of thrombotic events due to stasis of the left-ventricular blood flow, which intuitively suggests an increase of adverse effects with VA-ECMO treatment. However, in the only randomized controlled trial, the ECLS-SHOCK trial, VA-ECMO treatment did not impact negatively on left-ventricular function in 30 days survivors of acute myocardial infarction complicated with cardiogenic shock [29]. In addition, the favorable neurological outcome defined as a maximal modified Rankin-score ≤2 was more often in VA-ECMO-treated patients when compared to controls [30].

VA-ECMO treatment is therefore often applied in combination with left-ventricular (LV) decompression strategies such as IABP and IMPELLA treatment in order prevent LV dilatation resulting from increased LV afterload [31,32]. In fact, large meta-analyses suggest that LV decompression strategies either with IABP, IMPELLA, or left atrial or pulmonary artery cannulation is associated with lower mortality [31,33,34]. In accordance, LV decompression was likewise associated with a lower 30 day mortality in a propensity-matched cohort study while associated with an increased risk of severe bleeding, limb ischemia, and a need for renal replacement therapy [26,35]. Altogether, these results suggest a survival benefit when VA-ECMO treatment is associated with LV venting.

In summary, evidence from prospective randomized-controlled trials evaluating temporary MCS with conventional guideline-driven treatment is missing, but trials are ongoing in STEMI patients with concomitant cardiogenic shock (DOM Ger SHOCK, EUROSHOCK trial) [36,37] or cardiogenic shock without AMI (ECMO-CS, REVERSE) [38,39]. While these trials will not include all phenotypes of cardiogenic shock, their results will close, at least in part, this gap of knowledge.

## 4. Long-Term Left-Ventricular Mechanical Circulatory Support (LT-MCS)

Application of milrinone or dobutamine in advanced HF failed to show a decrease in morbidity or an increase in survival [40,41]. However, inotropic drug treatment may be an option in advanced HF if in accordance with the individual end-of-life care plan. Repetitive application of the inodilatator levosimendan in the outpatient setting is a promising candidate in this clinical setting [42], and the ongoing LEODOR trial (NCT03437226) compares this treatment with guidelines-directed medical therapy. While HTx still remains the most durable treatment option for eligible patients with advanced HF, the number of transplant operations remains limited to 4500 cases per year worldwide [43] and 35–50 cases per year in Switzerland [44]. HTx recipients are most often of younger age and without important comorbidity, whereas advanced HF patients are older and present with a more important comorbidity charge reason as to why they are often not considered to be suitable for HTx. This makes the case for LT-MCS implantation in the advanced HF patient who is refractory to optimal medical treatment and not eligible or opposed to HTx. This is more so true, since survival with modern continuous-flow left-ventricular assist-device (CF-LVAD) treatment has been shown to be superior to medical treatment alone in the ROADMAP-trial [45].

In theory, 10–25% of all advanced HF patients should qualify for LT-MCS treatment taking into account limitations related with age, comorbidity, or social constraint [46]. In reality, the annual implantation rate is substantially lower as indicated by a recent report from the United States [47]. On the basis of the latter report, an annual implantation rate ranging from to 100–250 may be expected in Switzerland, while, in reality, the annual implantation rate remains limited to 30–40 patients in the last years. This large difference is surprising and suggests that the medical community still is poorly familiar with the indication for LT-MCS and the clinical profile of a potential candidate for LT-MCS implantation.

As with other treatments, the best selection of the suitable candidate is primordial for LT-MCS treatment [48,49]. Therefore, the current European Society of Cardiology and European Association of Cardiothoracic Surgery guidelines define the indications for LVAD implantation not only as advanced systolic HF with left-ventricular ejection fraction (LVEF) <25% and NYHA functional class IIIb-IV despite optimal treatment. Candidates should likewise present with high one-year mortality as predicted by respective scores, by dependency on continuous intravenous inotropic support, or by fulfilling criteria indicating heart transplantation independent of whether destination therapy or HTx is the first intention [4,50].

While these specifications identify potential candidates for assist-device implantation, further stratification of advanced HF into seven different levels has proven useful for evaluating the urgency of LT-MCS implantation [51]. (Table 1) This stratification is nowadays endorsed by the Interagency Registry for Mechanically Assisted Circulatory Support (INTERMACS) and the European Association of Cardio-Thoracic Surgery (EACTS) and was applied for the recruitment of study participants in clinical trials testing continuous-flow LT-MCS treatment. The majority of advanced HF patients included in these studies were in INTERMACS levels 1–4; therefore, LT-MCS treatment in these patients is based on the largest available evidence. Set out on a clinically based subdivision of patients presenting with different severity grade of advanced HF, the INTERMACS stratification has already been shown useful for risk stratification when revealing that the INTERMACS level 1 is associated with a significantly worse outcome with immediate LT-MCS treatment when compared to other INTERMACS levels. In consistency with this finding, patients with INTERMACS level 1 are nowadays often bridged with temporary MCS towards candidacy to long-term assist-device implantation [50].

For the moment, assist-device treatment in Switzerland is largely reserved for HTx candidates worsening their clinical condition towards INTERMACS level 2–4, similar to other countries. However, the largest growth in LT-MCS treatment in the last years has been seen in advanced HF patients not suitable for HTx. These DT patients present a higher burden of comorbidity when compared to BTT patients, which can explain their higher mortality [52] as shown in the prospective MOMENTUM III trial. This landmark trial showed that HM3 device treatment in BTT vs. DT study participants (197 vs. 317 patients) was associated with a higher survival rate at 1 year and 2 years after implantation (1 year: 88.8 vs. 81.5%; 2 years: 76.8 vs. 73.2%; respectively) [53]. Corresponding survival data were reported from one larger Swiss CF-LT-MCS cohort where in BTT patients the 1 and 2 year mortality rates were 88.4% and 84.4% at 1 and 2 years, respectively [54], while a DT patients cohort reported 87.5 and 70% survival at 1 and 2 years, respectively (n = 16) [55].

With the growing experience and the progressively improving performance of LT-MCS, former limitations to LT-MCS such as age, BMI, frailty, or renal dysfunction are no longer considered strict contraindications to LT-MCS treatment. In particular, 14% of LT-MCS patients were >70 years old in the MCS Network Research, and their unadjusted survival was 75% and 65% at 1 and 2 years, respectively, while being 84% and 73% in younger patients [56]. However, survival was no longer different when renal function was normal in the elder CF-LVAD patients [57]; therefore, age > 70 years is no longer considered a strict limitation. Likewise, there is no cutoff of BMI above which LT-MCS is contraindicated since survival after assist device implant is not different between obese and non-obese patients, while HF readmission is more frequent in the former [58]. CF-LVAD implantation can even be applied as a BTC therapy in morbidly obese patients enabling recovery from obesity to a BMI < 35, which is the acknowledged upper limit for HTx [59]. In addition, frailty due to severe cardiac dysfunction does not longer present a contraindication since frailty may disappear with LT-MCS treatment if not resulting from cancer, lung disease, cirrhosis, liver disease, peripheral vascular, or neurological disease [60]. Renal dysfunction, although predictive for mortality after assist-device implantation [61], may also improve with LT-MCS treatment when secondary to reduced renal perfusion or an increase in right-atrial filling pressures. In contrast, structural kidney disease secondary to diabetes or hypertension is not likely to improve, but progression of renal dysfunction may slow with LT-MCS [50,62]. Furthermore, glycemic control may improve after restoration of normal cardiac output with LT-MCS [63], while severe end-organ damage from diabetes still remains a contraindication. Finally, preexisting pathology of coagulation may worsen after implantation of LT-MCS because of the fragmentation of the von Willebrand factor by device-related shear stress. Fortunately, the HeartMate 3 assist-device is associated with greater preservation of the macromolecular structure of the von Willebrand factor [64] suggesting that the incidence of bleeding may be lower with this device.

Despite all of this improvement, the right ventricle (RV) remains a point of concern since venous blood return will significantly increase after LT-MCS implantation, whereas unloading of the LV will induce a leftward septal shift. This results in an increase in the end-diastolic RV volume with a subsequent change in its geometry and will ultimately compromise RV function. These changes occur independent of preexisting RV function but are more likely to incidence when RV dysfunction preexists. RV dysfunction can also explain the higher incidence of postoperative bleeding, renal insufficiency, and prolonged length of hospital stay after LT-MCS implantation [65]. Preoperative identification of patients at high risk of postoperative RV failure is therefore essential, and advanced echocardiographic evaluation of the RV is recommended. However, preoperatively elevated central venous pressure (CVP) or CVP/PCWP ratio, severe renal dysfunction, and ventilator dependence are already fairly consistent predictors of severe right-ventricular failure after LT-MCS implantation [66]. Less-invasive surgery for LT-MCS placement when compared with conventional median sternotomy may be associated with a reduced incidence of RV failure after LT-MCS implantation as suggested from an international multicenter retrospective cohort [67]. This beneficial effect may relate to the fact that less-invasive surgery preserves pericardial constraint and, thus, better maintains LV twist and longitudinal septal motion [68]. However, this observation affords prospective testing in order to show whether this technique is an option for patients with a high risk for postoperative RV failure. In the future, robotic surgery may represent another noteworthy approach not only because it is a sternotomy sparing approach but also because it should maintain pericardial constraint. However, so far, only case-experience is reported [69,70].

In the past, more important RV dysfunction was considered a contraindication to LT-MCS. However, post-operative right-ventricular-acute MCS (RV-AMCS) devices have become more commonly used in the clinical setting of pre-operative RV dysfunction. Furthermore, designated algorithms help to select the proper device and provide recommendations for patient monitoring and weaning protocols [71]. In the case of unlikely recovery of RV dysfunction after LV LT-MCS implantation, a biventricular setup of continuous-flow VADs has been utilized. This therapeutic approach is complicated by a high rate of pump thrombosis [72]; therefore, implantation of a total artificial heart (THA) remains a valid option. Candidates for TAH are most often considered when HTx candidates suffer from acute irreversible biventricular failure at high risk for imminent death (INTERMACS class 1 or 2) and for whom a suitable donor is not available. The miniaturization of the console permits ambulation and aggressive physiotherapy, and this main benefit can explain why 60% of patients on TAH support at HTx present an overall 1-year survival posttransplant rate of 70% [73].

Finally, aortic-valve regurgitation affords careful evaluation since > mild aortic regurgitation mandates biological valve placement or application of a central leaflet coaptation stitch. However, both approaches increase the perioperative risk of RV dysfunction due to the need of extracorporeal circulation support in the operating room. However, >mild AR affords pre-implantation intervention because an increase in the severity of aortic regurgitation will result in a circulatory short-circuit between the CF-LVAD, the ascending aorta, and the left ventricle. Furthermore, a mechanic aortic prosthesis should be replaced by an aortic bioprosthesis, while a functional aortic bioprosthesis can remain in place [14].

In summary, the arrival of the last generation of continuous-flow LT-MCS together with the experience gained in the last years has expanded the pool of advanced HF patients who will benefit from LT-MCS treatment taking into account the remaining limitations (Figure 1).

## 5. Heart Transplantation

Heart transplantation (HTx) remains nonetheless the gold-standard treatment for advanced heart failure treatment despite the favorable results of durable MCS with recent continuous-flow LT-MCS [74]. The fact that the number of patients on the waiting list in the U.S. has increased by 37% in the years 2009 to 2018 may provide support to the statement that HTx remains nonetheless the superior choice for surgical treatment of advanced HF. However, this increase concomits with a progressive increase in the number of patients with cardiomyopathy of non-ischemic origin, while the number of patients with ischemic cardiomyopathy has remained largely unchanged [75]. Cardiomyopathy of non-ischemic origin often presents with right-ventricular dysfunction, which may prohibit left-ventricular LT-MCS [14], and this can explain the increase in HTx candidates on the U.S. waitlist. In contrast to the U.S., the numbers of patients on the waitlist in Switzerland have remained virtually unchanged from 2016 to 2020 [44]. Moreover, in a Swiss HTx center the number of patients with HTx for end-stage dilated cardiomyopathy of non-ischemic origin has decreased to 42.7%, while the number of end-stage HF patients due to ischemic cardiomyopathy has increased to 39.3% [76].

These antidromic developments may result from differences in the waitlist policy. In fact, the U.S. replaced more recently their long-standing three-tier system by a six-tier system based on the deliberation that increased granularity should improve waitlist grading of HF severity, and, thus, better identify the HTx candidate in need of urgent transplantation [77]. The waitlist policy has changed in Switzerland as well; however, the number of transplant operations in urgency status was sanctioned to an agreed maximum number of about 25% of all transplant operations, intending to reduce the disadvantage for patients on the normal waitlist. By this strategy, the number of HTx in urgency remained limited to 27–32% of all HTx operations in the years 2016–2020 [44] without an overall effect on one-year outcomes [76]. The six-tier system likewise did not affect one-year outcomes after HTx in the U.S. [78]; however, comparison on the basis of mortality alone remains insufficient since other parameters of post-transplant success such as time out of the hospital, freedom from dialysis, and improvement in physical and cognitive frailty are necessary to obtain a more-holistic view on the contemporary HTx waitlist policy.

Care for congenital heart disease (CHD) has significantly improved in the last decades with more than 90% of children born with these defects surviving until adulthood. HF is the leading cause of death in the majority of these adult patients, and more and more CHD are candidates for HTx [79] with beneficial long-term results [80]. However, LT-MCS is a valid option if a donor heart is not readily available [81,82].

Despite the clear need for an increased number of donor organs, the utilization rate of donor hearts has remained fairly low worldwide. The reasons for refusal of donor hearts in the U.S. have been a size mismatch, histo-incompatibility, older age, female donor gender, or donor-comorbidities [83]. Guidelines have traditionally recommended against donor undersizing by weight, especially for female donors [84]. Therefore, in an effort to improve size-matching metrics, various anthropometric parameters were tested for prediction of matching sorting heart mass as a superior predictor. In fact, undersizing of the donor organ based on heart mass matching was associated with increased mortality in a retrospective analysis of 19,168 HTx recipients in the UNOS data base (HR 1.34; 95% CI 1.13–1.59; *p* < 0.001) [85]. On the other hand, 32% of the donor hearts that were declined due to concerns of undersizing in the UNOS database of the years 2007 to 2016 were found eligible when using the heart mass as a parameter for matching metrics. While these results promise to improve the allocation of female donor hearts, a smaller study of 288 HTx recipients showed that undersizing of the donor heart affects early hemodynamics but was without impact on one-year survival [86]. Altogether, these results suggest that efforts to improve matching metrics have the potential to increase the allocation of donor organs judged not suitable based on current recommendations.

The number of HTx candidiates with allosensitization at a high titer of pre-formed HLA-antibodies has doubled over the last decades [43,87] and becomes more and more an obstacle to donor heart allocation. It affects between 6% to 9.2% of all HTx candidates, and while an earlier series of patients on LT-MCS reported a prevalence of allosensitization in up to two thirds of their patients [88], a contemporary series indicates a lower rate of 11% in patients on pulsatile LT-MCS and 15% in those on continuous-flow support [89]. However, allosensitization represents a threat to HTx programs since the number of BTT patients on the waitlist is increasing [76]. Furthermore, allosensitization decreases not only the potential donor pool for a given candidate but also prolongs the time on the waitlist. The allosensitized patient also faces an increased incidence of acute rejection and an overall worse survival posttransplant [90] despite the fact that the preformed antibodies at low-titer are considered safe for HTx in the immediate and early postoperative phase [91]. Traditional desensitization protocols as well as emerging therapies can be successfully trialed in the appropriate HTx candidate in order to reduce the number and the level of preformed antibodies and to increase transplant eligibility [92,93,94]. However, pretransplant desensitization protocols in the LT-MCS patient are not without risk particularly for cable infection [95]. Additionally, posttransplant strategies for prevention of humoral rejection may be beneficial in the short-term as shown more recently in a trial applying rituximab treatment within the first 12 postoperative days. However, rituximab application was associated in this trial with a marked increase in the atheroma burden in the cardiac allograft [96] questioning the mid-and long-term benefits of this treatment. However, a case series of HTx recipients with a humoral rejection posttransplant was shown to benefit from a complement blockade [97], suggesting that this therapeutic strategy may also apply for HTx of allosensitized patients.

Preservation and storage of the donor organ has become an important issue in today’s allocation since traveling distances are larger, prolonging cold ischemic times. In order to decrease myocardial damage from the unmet metabolic need during transport, a cardioplegic solution is applied at procurement to induce diastolic arrest. In addition, hypothermia during transport is maintained until reperfusion in the recipient [98]. Traditionally, hypothermia during transport is maintained by the use of ice externally cooling the pouch, which contains the donor heart in the transport medium [99]. However, an absent temperature control risks to expose, in particular, the thin-walled RV of the cardiac allograft to very low temperature conditions, which may result in primary graft dysfunction. In 2018, the Paragonix SherpaPak^TM^, which maintains a target temperature and avoids placing the donor organ in close proximity to ice, was presented [100]. Case reports suggest that the SherpaPak^TM^ enables longer cold ischemic transport times [101], however, controlled studies are mandated until broader application with a prolonged ischemia time can be recommended. Therefore, and until provision of further evidence, a maximal duration of 4 hours of cold ischemia time should be respected even in younger-age donors [102].

In recent years, continuous, normothermic perfusion with warm, oxygenated, nutrient-enriched donor blood provided by the TransMedics Organ Care System (OCS)^TM^ has been evaluated as another option for the transport of the donor organ to the recipient. The monitoring of the donor graft function is possible via visual inspection of the donor heart, measurement of the coronary blood flow, and serial arterial or venous lactate measurements. The PROCEED II trial (Randomized Study of Organ Care System Cardiac Preservation for Preservation of Donated Hearts for Eventual Transplantation) randomized 130 patients to standard cold storage or the OCS^TM^, and the results showed non-inferiority for the latter without a difference in 30 day patient and graft survival [103]. Of note, the total mean out-of-body time was longer, while the cold ischemia was shorter. This suggests that a longer transport time is feasible with the OCS^TM^ without increasing the risk of cardiac-related serious adverse events. The OCS has also been tested for the use of extended criteria donors in the OCS Heart EXPAND trial. Donor hearts in this study were of older donor age, presented left-ventricular dysfunction or hypertrophy, or were donor hearts after prolonged cardiac arrest, exposed to prior drug abuse, or presenting mild to moderate coronary disease. Application of the OCS resulted in an excellent short-term post-transplant survival (88% six-month survival) and low rates of primary graft dysfunction (10.7% rate of severe left- or right-ventricular dysfunction at 24 hours posttransplant) [104]. Another study compared the outcome of HTx with marginal donor hearts preserved by ex vivo normothermic perfusion (n = 26) or cold storage (n = 79). A total of 21/26 marginal donor hearts preserved by OCS^TM^ were successfully transplanted with 5/26 donor hearts discarded because considered unsuitable while on the OCS^TM^. These 21 patients presented an overall lower burden of postoperative complications when compared to control-group marginal donor hearts preserved by cold storage [105]. Altogether, the results from the latter studies suggest that ex vivo normothermic perfusion permits appraisal of marginal donor heart function and has the potential to exclude non-suitable marginal donor grafts from transplantation. Notably, marginal donor hearts considered suitable for transplantation on the OCS^TM^ had favorable short-term results after HTx. However, knowledgeable care of the ex vivo organ is primordial, and this is only provided with routine application of OCS^TM^-based preservation. In Switzerland, application of the OCS^TM^ faces financial limitations at the moment, especially because of expensive consumables. However, transport times up to 10 hours with successful subsequent HTx have been reported when donor hearts were procured and preserved using this technique [106,107]. Therefore, this system promises to provide solutions not only for procurement of donor hearts at large distance but also for the expansion of the donor heart pool.

In summary, a recent advance in the field of cardiac transplantation provides solutions for the growing demand for suitable organs. Improved size matching considering the heart mass has the potential to improve donor-recipient matching and to reduce the refusal rate based on mismatches of height, body weight, and gender, thereby expanding the donor pool. Furthermore, an advance in organ preservation permits not only procurement at larger distances but also ex vivo appraisal of the function of the donor heart meeting extended criteria for procurement with the potential to increase the number of donor hearts suitable for transplantation. The future of cardiac transplantation will rely on these advances so that a maximum of heart transplant recipients has the chance to derive the best quality of life and survival.

## 6. Taking Care of the Aged Advanced Heart Failure Patient in the French-Speaking Part of Switzerland

Since 2003, the University Hospitals of Geneva and Lausanne have collaborated in a common HTx program with the transplant operation and the immediate postoperative care provided by the latter hospital. In this context, the two centers have agreed on an algorithm paving the treatment of the advanced HF patient >65 years of age. Acknowledging that the indications for surgical care of the younger advanced HF patient are set out clearly by the International Society of Heart and Lung Transplantation and the European Association of Cardiothoracic Surgery [13,108], these were not taken into consideration by this algorithm (Figure 2).

The 10 year update of the listing criteria for HTx recommends that advanced HF patients can be considered for HTx if they are <71 years of age, while HTx remains an option in selected patients >70 years [108]. The overall consensus in Switzerland and in the present algorithm is that advanced HF patients can be waitlisted until the age of 65 years when meeting the listing criteria and if an absolute contraindication is absent. However, the option of HTx is only provided if the individual patient presents with INTERMACS class 5 to 7 promising survival until the transplant operation [4,108]. The Swiss consensus for HTx in advanced HF patients between 65–69 years old limits the option of HTx waitlisting to patients with cardiac monopathology and without larger additional comorbidity. In any case, urgent HTx is not an option when the clinical conditions of these patients decline (Figure 2). Furthermore, these HTx candidates are preferentially considered for extended-criteria donor hearts acknowledging that the one-year and five-year survival rates are not different if extended-criteria donor hearts are used as shown more recently in an analysis of the Scientific Registry of Transplant Recipients [109]. Therefore, these recipients are not competing with younger patients on the waitlist. In Switzerland, only 27.5% of all donor heart offers (n = 199/723) were transplanted in the years 2007–2013, and the mean age was 40.2 years [110], suggesting that application of extended criteria may expand the donor heart pool only by considering older-age donor hearts for transplantation. In addition, the results from the extended-criteria donor hearts evaluated by the OCS^TM^ indicate that expansion of the donor heart pool beyond older age and by inclusion of donor heart with structural pathology is possible [105].

Nonetheless, the usual option of surgical treatment of an advanced HF patient aged between 65–69 years is DT (Figure 2), especially if an advanced HF patient presents with INTERMACS class 3 and 4. The decision in favor of LT-MCS is furthermore fostered when the HeartMate 2 risk score [111], and evaluation of RV predicts a favorable outcome [66]. If these LT-MCS patients at the age of 65–69 years experience associated incurable complications such as recurrent episodes of ventricular storm or severe gastrointestinal bleeding, progressive aortic-valve insufficiency, or device-related severe infection, the algorithm provides the option for HTx but again without the possibility for urgent HTx. HTx is also not an option if LT-MCS patients are >70 years of age where optimal medical treatment or palliative treatment are foreseen in accordance with the end-of-life plan of the individual patient.

## 7. Conclusions

The arrival of the last generation of continuous-flow LT-MCS and the experience gained in the last years has increased the number of advanced HF patients eligible for LT-MCS treatment. This evolution is of importance in particular because of the progressively increasing number of advanced HF patients presenting with older age where HTx does remain an option for highly selected cases. However, the improvement in matching metrics between donors and recipients has the potential to expand the donor pool, resulting in transplantation for otherwise-refused donor hearts. In addition, recent development of organ preservation permits not only procurement at larger distances but also appraisal of extended-criteria donor hearts and, thus, has the potential to increase the number of donor hearts suitable for transplantation. Therefore, there is promise that more patients with advanced HF may have the benefit of surgical treatment of advanced HF already in the near future.

## Figures and Tables

**Figure 1 jcm-11-00773-f001:**
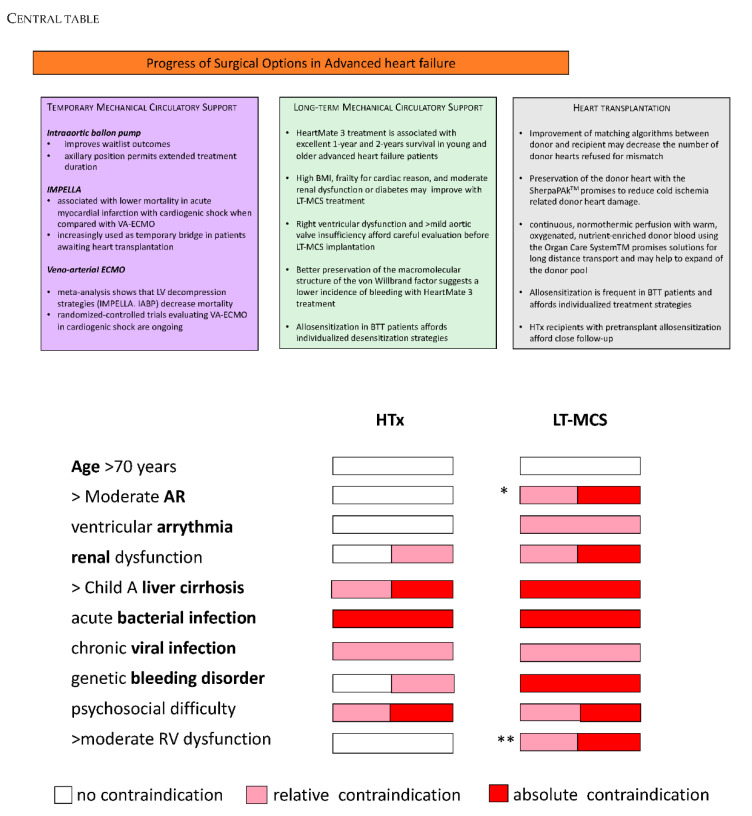
Indication, relative and absolute contraindications for temporary RV-AMCS, LT-MCS, and HTx. AR=aortic regurgitation; * mild AR = relative contraindication; >mild AR = absolute contraindication, aortic-valve replacement with bioprothesis is recommended; ** mild RV dysfunction = relative contraindication; >mild RV = consider perioperative strategies with LV or LT-MCS (see Kapur et al.; Farag et al.; Noly et al.); all indications or relative/absolute contraindications (see Costanzo M.R., Dipchand A., Starling R., et al. The International Society of Heart and Lung Transplantation Guidelines for the care of heart transplant *reciepients. J. Heart Lung. Transplant,*
**2010**, *29*, 915–955; *Gustafsson F, Rogers JG. Eur J Heart Fail*, **2017**, *19*, 595–602.

**Figure 2 jcm-11-00773-f002:**
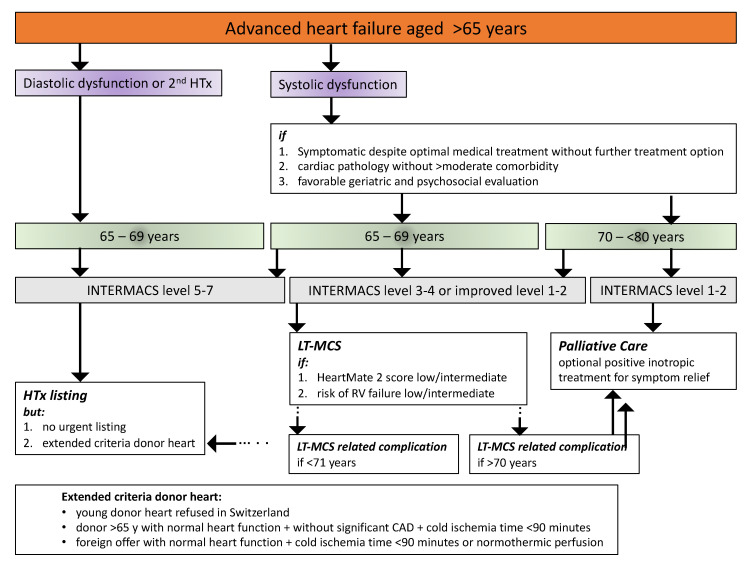
Algorithm describing the care of the aged patient with advanced heart failure in the French-speaking part of Switzerland.

**Table 1 jcm-11-00773-t001:** INTERMACS classification.

Level	Time to MCS
“crash and burn”: critical cardiogenic shock	within hours
“progressive decline”: inotrope dependence with progressive decline	within few days
“stable inotrope-dependent”: clinical stability with mild to moderate dose of intravenous inotropes or patients on temporary circulatory support without inotropes	within a few weeks
“recurrent advanced heart failure”: “recurrent” rather than “refractory” decompensation	within weeks to months
“exertion intolerant”: clinical stability, comfortable at rest but intolerant to exercise	variable
“exertion limited”: clinical stability, able to do mild activity but presentation of fatigue within a few minutes on any meaningful physical activity	variable
“advanced nyha 3”: clinical stability with a reasonable but variable level of physical activity and without recent decompensation	variable

## Data Availability

Not applicable.

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
