# Peer review of "Cardiac Surgery in Advanced Heart Failure"

_jcm, 2022, doi:10.3390/jcm11030773_

Round 1
Reviewer 1 Report
The authors clearly present current data on the treatment of the heart failure, with particular emphasis on surgical treatment. They discuss the limitations of the use of catecholamines in the treatment of chronic heart failure and present the possibilities of using MCS as a bridging therapy.
The easiest, fastest and cheapest to use as a temporary MCS is VA ECMO. However, the authors rightly note that only VA ECMO in combination with left ventricular decompression can give the expected results. These methods include LV vent, Impella or IABP.
The authors, using current publications, present the renaissance of the use of IABP in transplantology. However, they overlook the fact that not only Impella is currently used in acute conditions. According to current registries, IABP has several times wider application compared to microaxial LVAD in patients requiring temporary support during PCI. According to data from the US registry covering 28 304 patients undergoing PCI for AMI complicated by cardiogenic shock, an intravascular microaxial LVAD was used in 6.2% of patients, and IABP was used in 29.9%. (Sanket S. Dhruva, et al. : Association of Use of an Intravascular Microaxial Left Ventricular Assist Device vs Intra-aortic Balloon Pump With In-Hospital Mortality and Major Bleeding Among Patients With Acute Myocardial Infarction Complicated by Cardiogenic Shock. JAMA February 25, 2020 Volume 323, Number 8, 734-45.)
This phenomenon is also completely consistent with everyday cardiac surgical clinical practice, but it contradicts previous randomized trials, which are gradually reducing recommendations for IABP.
The authors point to much broader indications for LT-MCS than the actual application. The reason may be a limitation in the availability of the procedure related to the availability of equipment or the efficiency of the teams qualifying and performing this treatment. There may also be a lack of awareness of the medical community about the current indications for LT-MCS.
The authors emphasize that current recommendations precisely group patients to individual types of supportive therapy. The authors mention that currently in Switzerland MCS is mainly used as a bridge to HTx. This is a phenomenon observed in most countries.
The authors interestingly discuss many aspects of contraindications and limitations in the use of LT-MCS.
In Part II of the publication, the authors discuss individual aspects of HTx. Particular attention is paid to the fact of a constant increase in the number of patients waiting for transplantation. This is related to the increase in the number of patients with non-ischemical cardiomyopathy. In many countries, waitinglist systems are being modified.
The authors also discuss changes in organ procurement systems, matching, as well as methods of storing and transporting organs.
Author Response
Reviewer 1
The authors clearly present current data on the treatment of the heart failure, with particular emphasis on surgical treatment. They discuss the limitations of the use of catecholamines in the treatment of chronic heart failure and present the possibilities of using MCS as a bridging therapy.
The easiest, fastest and cheapest to use as a temporary MCS is VA ECMO. However, the authors rightly note that only VA ECMO in combination with left ventricular decompression can give the expected results. These methods include LV vent, Impella or IABP.
We thank the reviewer for this comment and refer to the lines 164-172: " ..., VA-ECMO treatment is often applied in combination with left ventricular (LV) decompression strategies such as IABP and IMPELLA treatment in order prevent LV dilatation resulting from increased left ventricular afterload [31,32]. In fact, large meta-analyses suggest that LV decompression strategies either with IABP, IMPELLA, or left atrial or pulmonary artery cannulation are associated with lower mortality [33,34]. In accordance, LV decompression was likewise associated with a lower 30-day mortality in a propensity-matched cohort study while associated with an increased risk of severe bleeding, limb ischemia, and need for renal replacement therapy [35]. Altogether, these results suggest a survival benefit when VA-ECMO treatment is associated with LV venting.
The authors, using current publications, present the renaissance of the use of IABP in transplantology. However, they overlook the fact that not only Impella is currently used in acute conditions.
We agree with the reviewer that not only Impella is currently used in acute conditions. The text discussed IABP treatment in lines 122-139 and only first evidence of the role of IMPELLA in lines 150-151.
According to current registries, IABP has several times wider application compared to microaxial LVAD in patients requiring temporary support during PCI. According to data from the US registry covering 28 304 patients undergoing PCI for AMI complicated by cardiogenic shock, an intravascular microaxial LVAD was used in 6.2% of patients, and IABP was used in 29.9%. (Sanket S. Dhruva, et al. : Association of Use of an Intravascular Microaxial Left Ventricular Assist Device vs Intra-aortic Balloon Pump With In-Hospital Mortality and Major Bleeding Among Patients With Acute Myocardial Infarction Complicated by Cardiogenic Shock. JAMA February 25, 2020 Volume 323, Number 8, 734-45.) This phenomenon is also completely consistent with everyday cardiac surgical clinical practice, but it contradicts previous randomized trials, which are gradually reducing recommendations for IABP.
We thank the reviewer for this comment and added this information in the manuscript (lines 121-122)." Nonetheless, IABP is still broadly applied in this clinical setting [20]."
The authors point to much broader indications for LT-MCS than the actual application. The reason may be a limitation in the availability of the procedure related to the availability of equipment or the efficiency of the teams qualifying and performing this treatment. There may also be a lack of awareness of the medical community about the current indications for LT-MCS.
We agree with the reviewer and this consideration is now provided in the manuscript (lines 201-203). " This large difference is surprising and suggests that medical community still is poorly familiar with the indication for LT-MCS and the clinical profile of a potential candidate."
The authors emphasize that current recommendations precisely group patients to individual types of supportive therapy. The authors mention that currently in Switzerland MCS is mainly used as a bridge to HTx. This is a phenomenon observed in most countries.
The reviewer is right, therefore, we added in the manuscript lines 226-228." For the moment, assist device treatment in Switzerland is largely reserved for HTx candidates worsening their clinical condition towards INTERMACS level 2-4 similar to other countries."
The authors interestingly discuss many aspects of contraindications and limitations in the use of LT-MCS.
We thank the reviewer for this grateful comment.
In Part II of the publication, the authors discuss individual aspects of HTx. Particular attention is paid to the fact of a constant increase in the number of patients waiting for transplantation. This is related to the increase in the number of patients with non-ischemical cardiomyopathy. In many countries, waitinglist systems are being modified.
We agree with the reviewer. Waiting list policy is not published for most countries. Therefore, we decided to refer to the U.S. waitlist policy, which is published, and the Swiss policy that we are working with.
This is a potentially noteworthy contribution in the context of a topical issue; however, I believe that the following points should be addressed before consideration for acceptance and publication.
We thank the reviewer for the grateful comments.
Reviewer 2 Report
In their manuscript, the Authors comprehensively summarized the recent advances in the surgical management of patients with advanced heart failure.
This is a potentially noteworthy contribution in the context of a topical issue; however, I believe that the following points should be addressed before consideration for acceptance and publication.
- The Authors should discuss the current evidence about the mechanical circulatory support devices for right ventricular failure, total artificial heart and biventricular assist devices (Kapur NK, Esposito ML, Bader Y, Morine KJ, Kiernan MS, Pham DT, Burkhoff D. Mechanical Circulatory Support Devices for Acute Right Ventricular Failure. Circulation. 2017 Jul 18;136(3):314-326. doi: 10.1161/CIRCULATIONAHA.116.025290. PMID: 28716832; Pierre-Emmanuel Noly, Wallid Ben Ali, Yoan Lamarche, Michel Carrier, Status, Indications, and Use of Cardiac Replacement Therapy in the Era of Multimodal Mechanical Approaches to Circulatory Support: A Scoping Review, Canadian Journal of Cardiology, Volume 36, Issue 2, 2020, Pages 261-269, ISSN 0828-282X, https://doi.org/10.1016/j.cjca.2019.11.027; Farag J, Woldendorp K, McNamara N, Bannon PG, Marasco SF, Loforte A, Potapov EV. Contemporary outcomes of continuous-flow biventricular assist devices. Ann Cardiothorac Surg. 2021 May;10(3):311-328. doi: 10.21037/acs-2021-cfmcs-34. PMID: 34159113; PMCID: PMC8185384).
- The Authors should summarize in a table the indications and contraindications regarding the use of temporary mechanical circulatory support and long term left ventricular mechanical circulatory support as well as candidacy to heart transplantation.
- A brief summary about the surgical management of advanced heart failure in the pediatric population and in congenital heart diseases should be provided (Menachem JN, Schlendorf KH, Mazurek JA, Bichell DP, Brinkley DM, Frischhertz BP, Mettler BA, Shah AS, Zalawadiya S, Book W, Lindenfeld J. Advanced Heart Failure in Adults With Congenital Heart Disease. JACC Heart Fail. 2020 Feb;8(2):87-99. doi: 10.1016/j.jchf.2019.08.012. Epub 2019 Dec 11. PMID: 31838031; Monda E, Lioncino M, Pacileo R, Rubino M, Cirillo A, Fusco A, Esposito A, Verrillo F, Di Fraia F, Mauriello A, Tessitore V, Caiazza M, Cesaro A, Calabrò P, Russo MG, Limongelli G. Advanced Heart Failure in Special Population-Pediatric Age. Heart Fail Clin. 2021 Oct;17(4):673-683. doi: 10.1016/j.hfc.2021.05.011. Epub 2021 Jul 22. PMID: 34511214).
Please, use full names of abbreviations when used for the first time (p.e. MCS).
Author Response
Reviewer 2
The Authors should discuss
the current evidence about the mechanical circulatory support devices for right ventricular failure, total artificial heart and biventricular assist devices (Kapur NK, Esposito ML, Bader Y, Morine KJ, Kiernan MS, Pham DT, Burkhoff D. Mechanical Circulatory Support Devices for Acute Right Ventricular Failure. Circulation. 2017 Jul 18;136(3):314-326. doi: 10.1161/CIRCULATIONAHA.116.025290. PMID: 28716832;
We thank the reviewer for this citation and integrated the information provided with this publication in the revised version of the manuscript (lines 276-280). " In the past, more important RV dysfunction was considered a contraindication to LT-MCS. However, post-operative RV-AMCS (right-ventricular-acute MCS) devices become more commonly used in the clinical setting of pre-operative RV dysfunction and designated algorithms help to select the proper device, and provide recommendations for patient monitoring and weaning protocols [Kapur et al.]."
Pierre-Emmanuel Noly, Wallid Ben Ali, Yoan Lamarche, Michel Carrier, Status, Indications, and Use of Cardiac Replacement Therapy in the Era of Multimodal Mechanical Approaches to Circulatory Support: A Scoping Review, Canadian Journal of Cardiology, Volume 36, Issue 2, 2020, Pages 261-269, ISSN 0828-282X, https://doi.org/10.1016/j.cjca.2019.11.027;
We thank the reviewer for this citation and integrated the information provided with this publication in the revised version of the manuscript. Line 284-289: "Candidates for TAH are most often considered when HTx candidates suffer from acute irreversible biventricular failure at high risk for imminent death (INTERMACS class 1 or 2) and for whom a suitable donor is not available. The miniaturization of the console permits ambulation and aggressive physiotherapy, and this main benefit can explain why the 60% of patients on TAH support at HTx present an overall 1-year survival posttransplant of 70% (Noly et al.)."
Farag J, Woldendorp K, McNamara N, Bannon PG, Marasco SF, Loforte A, Potapov EV. Contemporary outcomes of continuous-flow biventricular assist devices. Ann Cardiothorac Surg. 2021 May;10(3):311-328. doi: 10.21037/acs-2021-cfmcs-34. PMID: 34159113; PMCID: PMC8185384).
We thank the reviewer for this citation and integrated the information provided with this publication in the revised version of the manuscript (lines 280-283): " In the case of unlikely recovery of RV dysfunction after LV LT-MCS implantation, a biventricular setup of continuous-flow VADs has been utilized. This therapeutic approach is complicated by a high rate of pump thrombosis (Farag et al.), therefore, implantation of a total artificial heart (THA) remains a valid option."
The Authors should summarize in a table the indications and contraindications regarding the use of temporary mechanical circulatory support and long term left ventricular mechanical circulatory support as well as candidacy to heart transplantation.
We thank the reviewer for this recommendation and created table 2.
A brief summary about the surgical management of advanced heart failure in the pediatric population and in congenital heart diseases should be provided (Menachem JN, Schlendorf KH, Mazurek JA, Bichell DP, Brinkley DM, Frischhertz BP, Mettler BA, Shah AS, Zalawadiya S, Book W, Lindenfeld J. Advanced Heart Failure in Adults With Congenital Heart Disease. JACC Heart Fail. 2020 Feb;8(2):87-99. doi: 10.1016/j.jchf.2019.08.012. Epub 2019 Dec 11. PMID: 31838031;
Monda E, Lioncino M, Pacileo R, Rubino M, Cirillo A, Fusco A, Esposito A, Verrillo F, Di Fraia F, Mauriello A, Tessitore V, Caiazza M, Cesaro A, Calabrò P, Russo MG, Limongelli G. Advanced Heart Failure in Special Population-Pediatric Age. Heart Fail Clin. 2021 Oct;17(4):673-683. doi: 10.1016/j.hfc.2021.05.011. Epub 2021 Jul 22. PMID: 34511214).
We agree with the reviewer that surgical treatment of pediatric cases with advanced heart failure is relevant and therefore added in lines 329-333: "Care for congenital heart disease (CHD) has significantly improved in the last decades with more than 90% of children born with these defects surviving until adulthood. HF is the leading cause of death in the majority of these adult patients and more and more CHD are candidates for HTx (Menachem et al.) with beneficial long-term results (Lund et al.). However, LT-MCS is a valid option if a donor heart is not readily available (Cedars et al., Monda et al.)."